# Effects of Mangosteen (*Garcinia mangostana*) Peel Extract Loaded in Nanoemulsion on Growth Performance, Immune Response, and Disease Resistance of Nile Tilapia (*Oreochromis niloticus*) against *Aeromonas veronii* Infection

**DOI:** 10.3390/ani13111798

**Published:** 2023-05-29

**Authors:** Jakarwan Yostawonkul, Manoj Tukaram Kamble, Kitikarn Sakuna, Sulaiman Madyod, Pimwarang Sukkarun, Seema Vijay Medhe, Channarong Rodkhum, Nopadon Pirarat, Mariya Sewaka

**Affiliations:** 1International Graduate Course of Veterinary Science and Technology (VST), Faculty of Veterinary Science, Chulalongkorn University, Bangkok 10330, Thailand; jo_jak49@hotmail.com; 2Wildlife, Exotic and Aquatic Animal Pathology Research Unit, Department of Pathology, Faculty of Veterinary Science, Chulalongkorn University, Bangkok 10330, Thailand; maav.manya@gmail.com (M.T.K.); seemamedhe@gmail.com (S.V.M.); 3Faculty of Veterinary Science, Rajamangala University of Technology Srivijaya, Nakhon Si Thammarat 80240, Thailand; kitikarn.s@rmutsv.ac.th (K.S.); sulaiman.m@rmutsv.ac.th (S.M.); pimwarang.s@gmail.com (P.S.); 4Center of Excellence in Fish Diseases (CE FID), Department of Veterinary Microbiology, Faculty of Veterinary Science, Chulalongkorn University, Bangkok 10330, Thailand; channarong.r@chula.ac.th

**Keywords:** *Aeromonas veronii*, mangosteen peel extract, nanoemulsion, antimicrobial, growth performance, immune response, disease resistance, Nile tilapia

## Abstract

**Simple Summary:**

Natural infections caused by *Aeromonas veronii* in intensive farming can lead to economic losses in tilapia farming. Overusing antibiotics and chemical antimicrobial agents in fish farming leads to antibiotic resistance, pollution, and consumer reluctance. The utilization of mangosteen (*Garcinia mangostana*) peel extract loaded in nanoemulsion (MSNE)-supplemented diets in Nile tilapia (*Oreochromis niloticus*) could improve growth performance, immune response, and disease resistance. Nevertheless, the effect of incorporating MSNE into Nile tilapia diets has not yet been studied. In this study, we assessed the efficacy of MSNE-supplemented diets on growth performance, immune response, and resistance to *A. veronii* infection in Nile tilapia. The particle size, polydispersity index, and particle surface charge of MSNE were 151.9 ± 1.4 nm, >0.3, and −30 mV, respectively. Furthermore, MSNE improved the in vitro inhibition against *A. veronii,* and MSNE-supplemented diets had a beneficial effect on growth performance, enhanced immune response, and disease resistance against *A. veronii* infection. In conclusion, mangosteen peel extract loaded in nanoemulsion has the potential to be used as a supplement in Nile tilapia culture.

**Abstract:**

Nanotechnology can enhance nutrient delivery and bioavailability; hence, it has recently been considered the most practical alternative technology for nutritional supplements and disease control in fish farming. The present study was designed to evaluate the effects of mangosteen peel extract loaded in nanoemulsion (MSNE) on the inhibition of *A. veronii* (in vitro) and in vivo growth performance, serum biochemical parameters, the immune response, and the disease resistance of Nile tilapia (*Oreochromis niloticus*) against *A. veronii* challenge. The particle size, polydispersity index, and particle surface charge of MSNE were 151.9 ± 1.4 nm, >0.3, and −30 mV, respectively. Furthermore, MSNE, mangosteen peel extract (MPE), and nanoemulsion (NE) improved the antimicrobial activity against *A. veronii*. Fish fed MSNE, MPE, and NE-supplemented diets had a significantly lower (*p* < 0.05) feed conversion ratio (FCR) and higher specific growth rate (SGR) than fish fed the control diet. Furthermore, the MSNE had significantly higher serum glucose and protein levels than the control group in Nile tilapia. Total immunoglobulin, serum lysozyme, alternative complement activity, and survival of Nile tilapia fed with MSNE were significantly higher (*p* < 0.05) than the control diet. Therefore, MSNE has the potential to be employed as a supplement in sustainable Nile tilapia farming.

## 1. Introduction

Nile tilapia (*Oreochromis niloticus*) is a popular farmed fish in Thailand due to its faster growth, good flavor, and disease resistance [1,2,3]. The global production of Nile tilapia in 2020 was 4407.2 thousand tons [4], with Thailand contributing 210,419 tons, with a 2.2% increase in 2021 [5]. Nevertheless, pathogen infections in intensive tilapia farming have been a significant cause of economic losses in fish farming. Natural pathogen infections in tilapia have been found, such as those caused by *Aeromonas* spp., *Flavobacterium columnare*, *Edwardsiella* spp., *Francisella* spp., and *Streptococcus agalactiae*. Among bacterial diseases, *A. veronii* has recently resulted in high mortality rates on Thai tilapia farms [6,7]. The clinical signs of *A. veronii* infection in tilapia included ulceration, hemorrhagic septicemia, and enteritis, which are more common in juvenile tilapia because of their weaker immunity than adult tilapia [6,7,8].

In tilapia farms, antibiotics and chemotherapeutics have been used to treat the disease. On the other hand, the overuse of antibiotics and chemical antimicrobial agents leads to antibiotic resistance, fish farming pollution, and consumer reluctance [9,10]. In recent years, plant extracts [11,12,13], probiotics, and prebiotics [14,15] have been considered as alternatives for treating diseases in Nile tilapia caused by *A. veronii*.

Mangosteen (*G. mangostana*) is known as the “queen of fruits” and one of the best-tasting tropical fruits, which are consumed fresh or processed into jam, preserves, and wine [16]. However, the processing of mangosteen food products turns the mangosteen peel into waste. The mangosteen peel has been reported to contain bioactive compounds such as xanthones, tannins, phenolic acids, and other bioactive compounds [17]. Importantly, the mangosteen peel contains 40% xanthones (which are α-mangostins, β-mangostins, γ-mangostins, garcinone E, 8-deoxygartanin, and gartanin), which possess antimicrobial, antioxidant, anti-inflammatory, and anti-proliferative activities [18]. Moreover, mangosteen peel extract has been used to improve the survival of fish. Previous studies showed that mangosteen peel extract inhibits *F. columnare* in channel catfish (*Ictalurus punctatus)* [19] and increases the survival of black tilapia (*O. niloticus* Bleeker) against *A. hydrophila* [20]. Furthermore, a rind powder and shoot extract of mangosteen supplemented diet improved the hematological profile of clown anemonefish (*Amphiprion percula*) [21] and African catfish (*Clarias gariepinus*) fingerlings [22].

In aquaculture, nano-dietary delivery is a novel method to improve the bioavailability and efficacy of nutraceuticals. This is because nano-dietary delivery ensures that the targeted elements reach the bloodstream more effectively [23,24]. However, mangosteen peel extract provides low biological activity owing to its low water solubility and ease of decomposition. To overcome this challenge, nanoemulsions have been developed and used to deliver hydrophobic mangosteen peel extract. Nanoemulsion is one of the most important types of emulsion, which consists of ultra-fine particles in the range of 20–200 nm [25]. Oil-in-water (o/w) nanoemulsions are a specific type of nanoemulsion that has been utilized to deliver hydrophobic compounds. Furthermore, the hydrophobic drug, which was loaded into a nanoemulsion, could improve biological activity, promote active compound stability, and increase drug absorptivity in their target organ. Nanoemulsions are typically synthesized from two immiscible solutions: oil and an aqueous solution. An appropriate amount of surfactant and energy is added to the oil–aqueous mixture to form a nanoemulsion, which can be synthesized using various techniques, such as high- and low-energy homogenization, based on its chemical composition and is easy to scale up [26]. Importantly, the effect of incorporating mangosteen peel extract loaded in nanoemulsion into Nile tilapia diets has not been studied yet. Therefore, the aim of the present study was to evaluate the antimicrobial efficacy of mangosteen (*G. mangostana*) peel extract loaded in nanoemulsion and their supplemented diets on the growth performance, serum biochemical parameters, immune response, and disease resistance of Nile tilapia against *A. veronii* infection.

## 2. Materials and Methods

### 2.1. Preparation of Mangosteen Peel Extract Loaded Nanoemulsion

Mangosteen peels were obtained from the orchard in Lan Saka district, Nakhon Si Thammarat province, Thailand. The samples were cleaned, sliced into thin pieces, dried at 60 °C for 72 h, and ground into powder using a hammer grinder. The mangosteen peel extract (MPE) was extracted with 95% ethanol (*w*/*v*) at room temperature for 48 h, followed by filtering and evaporation using a rotary evaporator (Buchi, Switzerland). The ethanol extract of mangosteen peels is composed of xanthones [27,28], which possess antimicrobial, antioxidant, anti-inflammatory, and anti-proliferative activities [18].

Mangosteen peel extract loaded into a nanoemulsion was prepared for the supplemented diets. Briefly, mangosteen peel extract loaded into nanoemulsion (MSNE) was fabricated using hot and high homogenization energy. The oil phase was prepared by mixing mangosteen peel extract (200 mg) with medium-chain triglyceride (MCT) as a liquid lipid (15 g) and cetyl palmitate (5 g). The oil mixture was dissolved with Span 80 (3 g) and Montanov 82 (1 g) over a hotplate stirrer at 500 rpm and 60–70 °C. Over a hotplate stirrer, purified water, Tween 20 (3 g), glycerol (2.5 g), and synperonic PE/F68 (2 g) were mixed to make an aqueous phase mixture. Additionally, this mixture was poured into the lipid phase and sonicated for 5 min in the sonicator unit (Qsonica sonicator, Newtown, CT, USA) using a 40-amp pulse on for 30 s and off for 5 s intervals. For the supplemented diets, a nanoemulsion solution (NE) was prepared using chemicals similar to those used in MSNE but without mangosteen peel extract.

### 2.2. Characterizations of MSNE

The hydrodynamic diameter, polydispersity index (PDI), and particle surface charge of MSNE were characterized with dynamic light scattering (DLS) using a zetasizer (Nano ZS, Malvern Instrument, Malvern, Worcs, UK). DLS measurements were carried out using a He-Ne laser (λ0 = 633 nm, θ = 173°). The samples were diluted 20 times with purified water. The measurement conditions were set and performed in triplicate at 25 °C.

Particle size and particle morphology were also observed using a transmission electron microscope (TEM) (JEOL-2100 Plus, JEOL, Akishima, TYO, Japan). The samples were also diluted 1/50 in purified water and dropped onto a carbon grid. The prepared samples were dried in a dry cabinet overnight before being characterized. The samples were observed under 80 kV with a magnification of ×25 k and ×100 k.

### 2.3. Antibacterial Activity

#### 2.3.1. Broth Microdilution Assay

The minimum inhibitory concentration (MIC) and the minimum bactericidal concentration (MBC) were evaluated using the broth microdilution assay [29]. Briefly, *A. veronii* was isolated from Nile tilapia (*O. niloticus*) farming in Nong Khai province, northeastern Thailand [6]. Furthermore, *A. veronii* was cultured overnight at 30 °C on trypticase soy broth (TSB). Two-fold serial dilutions from a stock solution (100 mg of peel extract mL^−1^) of MPE, MSNE, and enrofloxacin were prepared (each with three replicates) in Mueller–Hinton broth (MHB). Furthermore, 10^8^ cells mL^−1^ (adjusted using 0.5 McFarland standard) of *A. veronii* were added to the solution, incubated at 30 °C for 24 h, and the growth was measured using a spectrophotometer at a wavelength of 625 nm. The MIC values were determined as the lowest concentration of compounds whose absorbance was comparable with the negative control tubes (MHB without inoculums). The minimum bactericidal concentration (MBC) was measured by culturing all of the tubes without turbidity. The MBC value is the lowest concentration of compounds and does not reflect bacterial growth.

#### 2.3.2. Disc Diffusion Assay

Antibacterial activity was determined by using a disc diffusion assay [30]. Briefly, *A*. *veronii* was cultured overnight at 30 °C on trypticase soy broth (TSB). The bacterial density (10^8^ cells mL^−1^) was inoculated on Mueller–Hinton agar (MHA) [31]. The MPE (62.5 µL mL^−1^), MSNE (62.5 µL mL^−1^), and NE (62.5 µL mL^−1^) were filtered (0.22 µm pore size), and 50 µL supernatants were placed on sterile paper discs (diameter 6 mm) on MHA. The plates were incubated at 30 °C for 24 h. The antibacterial activity was assessed by measuring the inhibition zone. Enrofloxacin 5 µg was used as a positive control, and dimethyl sulfoxide (DMSO, 0.5%) was used as a negative control [32,33].

### 2.4. Experimental Fish

Three hundred sixty monosex (male) tilapia (15.35 ± 0.91 g) were obtained from the Napho Phanpla Limited Partnership tilapia farm in Nakhon Si Thammarat, Thailand. The fish were divided into twelve 500-L tanks (30 each) and allowed to acclimatize for two weeks. Water temperature, dissolved oxygen, and pH levels were maintained between 25–28 °C, 5.24–5.98 mg L^−1^, and 7.48–8.16, respectively. Ammonia nitrogen was managed by exchanging 50% of the water every two days and measuring the ammonia nitrogen using test kits (V-Unique, Bangkok, Thailand), which indicated less than 0.02 mg L^−1^. The fish were hand-fed approximately 5% of their body weight twice a day. All of the protocols were approved by the ethics committee of Rajamangala University of Technology Srivijaya (Approval No. U1-03662-2559).

The in vivo concentration was selected based on the results of the MIC and inhibition zone. The complete randomized design with four treatments was carried out in triplicate. The stock solution of extract, i.e., 100 mg mL^−1^ distilled water, was used to prepare the experimental diets. The treatment diets were as follows: a control diet (Control), MPE (62.5 µL mL^−1^, i.e., 6.25 mg g^−1^ of feed), MSNE (62.5 µL mL^−1^, i.e., 6.25 mg g^−1^ of feed), and NE (62.5 µL mL^−1^). The control and the experimental diets were prepared by thoroughly mixing 1 mL of the MPE, MSNE, or NE, and 1 mL of distilled water (control) with 1 g of feed in the commercial diet (Charoen Pokphand Foods Public Company Limited, Samut Sakhon, Thailand), air-dried, and 4 °C used to store until feeding. The proximate composition of the commercial feed was as follows: lipid (3%), protein (30%), moisture (12%), and ash (8%).

### 2.5. Growth Performance

After 30 days of feeding, the standard formulas were used to calculate the growth performance and feed utilization performance of Nile tilapia [34].
Weight gain (WG) (g fish^−1^) = (final body weight (FW) − initial body weight (IW))(1)
Specific growth rate (SGR) = [(ln (FW) − ln (IW)/days] × 100(2)
Feed conversion ratio (FCR) = feed intake (g)/WG(3)
Average daily gain (ADG) = (% gain)/(number of days)(4)

### 2.6. Serum Biochemical Analysis

After 30 days of feeding, blood samples (6 fish per group) were collected from the caudal vein using a hypodermic syringe. The blood samples were allowed to clot for 3 h at 4 °C, and the serum was collected after centrifugation at 2600× *g* for 10 min at room temperature. The serum samples were used to measure blood urea nitrogen (BUN), total protein, glucose, albumin, direct bilirubin (D-bilirubin), total bilirubin (T-bilirubin), serum aspartate aminotransferase (AST), serum alanine transaminase (ALT), and total cholesterol using an automated chemistry analyzer (Pokleritalia 125, PKL, Italy).

### 2.7. Immunological Assay

The total immunoglobulin (Ig) was estimated by using the method of Siwicki et al. [35]. The total plasma protein concentration was determined with bovine serum albumin (standard protein). The plasma protein was precipitated with 12% polyethylene glycol, incubated at room temperature for 30 min, and centrifuged at 12,500 rpm for 10 min. The supernatant (10 µL) was mixed with 500 µL of biuret reagent. It was incubated for 5 min at room temperature, and the absorbance was measured using a spectrophotometer at a wavelength of 550 nm. The precipitation of plasma protein concentration was determined with bovine serum albumin (standard protein). The total immunoglobulin was calculated by subtracting the precipitation of plasma protein concentration from the total plasma protein concentration.

The alternative complement hemolytic 50 (ACH50) activity was analyzed [36]. Briefly, the serum was diluted in GVB-EGTA (gelatin Veronal buffer; 10 mM barbital, 145 mM NaCl, 0.1% gelatin, 0.5 mM MgCl_2_, 10 mM EGTA, pH 7.3–7.4) to a final volume of 250 µL. Then, 50 µL of goat red blood cells was added to the test serum for the preparation of a 2-fold serial dilution and incubated at room temperature for 90 min. The relative hemoglobin content of the supernatant was assessed using a spectrophotometer at a wavelength of 415 nm. The ACH50 activity was determined by assessing the amount of serum that induces 50% lysis of goat red blood cells.

The activity of lysozyme in serum was evaluated by indicating the level of lysis of the Gram-positive bacterium *Micrococcus luteus*. Briefly, the lysozyme standard was diluted in 0.06 M phosphate citrate buffer (pH 6.0) and 0.09% NaCl to concentrations of 0, 2.5, 5, 10, 15, and 20 µg mL^−1^. The 100 µL of the lysozyme standard and the serum were added to 96 microplates with the addition of *M. luteus*. The absorbance was measured using a spectrophotometer at a wavelength of 450 nm [37].

### 2.8. Challenged Study

At the end of the experimental period (30 days), thirty fish from each group were challenged with intraperitoneal injection with *A. veronii* at 10^7^ CFU fish^−1^, based on the previous study [6]. Afterward, mortalities or any clinical signs were observed for 15 days. The survival rate (SR) was calculated as follows [2]:SR (%) = (Total no. of survivors after challenge/total number of fish challenge) × 100(5)

In addition, the relative percentage of survival (RPS) was calculated as follows [15]
RPS = [1 − [(treatment mortality/control mortality) × 100](6)

### 2.9. Statistical Analysis

Statistical analysis was conducted by using SPSS version 26 software for Windows (SPSS Inc., Chicago, IL, USA). The results were analyzed using a one-way analysis of variance (ANOVA), and significant differences between the groups were determined through the use of Duncan′s multiple range tests. The cumulative survival percentages of the experimental groups were analyzed using the Kaplan–Meier method and the Log Rank (Mantel-Cox) test. A difference of *p* < 0.05 was considered significant.

## 3. Results

### 3.1. Characterizations of MSNE

The particle sizes of MSNE and NE were 151.9 ± 1.4 nm and 146.4 ± 3.1 nm, respectively (Table 1). The polydispersity index and particle surface charge of MSNE and NE were lower than 0.3 and −30 mV, respectively. These results suggested that the protocol was successful in preparing nanoparticles.

TEM results reported that the morphology of MSNE was spherical, and the particle size was smaller than 200 nm (Figure 1).

### 3.2. Antibacterial Activity of MSNE

Both the minimum inhibitory concentration (MIC) and minimum bactericidal concentration (MBC) of MPE and MSNE were 62.5 µL mL^−1^. The results of the disc diffusion assay showed that the inhibition zone for *A. veronii* in MSNE (16.00 ± 3.00 mm) was significantly higher (*p* < 0.05) than the MPE (11.67 ± 1.15 mm) and NE (10.67 ± 1.15 mm) (Table 2 and Figure 2). MSNE, on the other hand, had a significantly lower (*p* < 0.05) inhibition zone than enrofloxacin (24.67 ± 1.15 mm).

### 3.3. Growth Performance of MSNE-Supplemented Diet

Fish fed with the MSNE diet had a significantly lower FCR (*p* < 0.05) than fish fed with the control, MPE, and NE diets, respectively (Table 3). In addition, the SGR, WG, and ADG of the MSNE diet were significantly (*p* < 0.05) higher than the control diet. Furthermore, MPE and NE diets also found higher WG and ADG compared to the control. The results indicate that the MSNE diet did not have any detrimental effect on Nile tilapia.

### 3.4. Serum Biochemical Analysis of MSNE-Supplemented Diet

The MSNE diet had significantly higher glucose and total protein levels than the control group (Table 4). Moreover, the results revealed that there were no significant differences in BUN, ALT, AST, total bilirubin, direct bilirubin, total cholesterol, or albumin among the groups.

### 3.5. Immune Parameters Analysis of MSNE-Supplemented Diet

The fish fed with MSNE had a significant increase (*p* < 0.05) in total Ig in comparison with the NE and control groups, respectively (Figure 3). The lysozyme activity and ACH50 activity of MSNE, MPE, and NE were significantly different in fish fed with the control diets.

### 3.6. Survival Rates of MSNE-Supplemented Diet

The mortality in the control group (66.7%), MPE (56.7%), MSNE (40.0%), and NE (66.7%) occurred on day 2 post-challenge. Importantly, fish mortality stopped at day 7, day 5, day 4, and day 7 in the control, MPE, MSNE, and NE, respectively (Figure 4A). A log-rank test showed that the survival percentages of the four experimental groups were significantly different (*X*^2^(3) = 6.893, *p* < 0.05) (Figure 4A). All of the fish deaths were caused by *A. veronii*, as determined through bacterial isolation from the spleen and liver. All of the dead fish showed clinical signs of a pale body surface, hemorrhage in the liver, and a swollen intestine with an accumulation of yellow liquid (Figure 4B,C).

The cumulative mortality was significantly lower in the MSNE (76.7%) and MPE (60%) diets than in the control and NE groups (Table 5). The relative percent of survival (RPS) of the Nile tilapia fed MSNE (39.1%) was significantly higher than those of MPE (21.7%), and NE (4.3%).

## 4. Discussion

Nanotechnology improves nutrient delivery by increasing solubility and protecting against the harsh conditions of the gut, resulting in increased fish potential for nutrient absorption [38,39]. Importantly, nanotechnology has the potential to become a common practical nutritional supplement and fish disease control technology in fish farming [39]. Therefore, the current study evaluated the efficacy of supplementing Nile tilapia diets with mangosteen peel extract loaded in nanoemulsion (MSNE).

In the present study, the particle sizes and zeta potential of MSNE were 151.9 ± 1.4 nm and −30 mV, confirming the successful formation of nanoemulsions that have a droplet size ranging from 10 to 500 nm [40]. The mean particle size and the negatively charged surface result are in accordance with the previous study on the nanoemulsion of mangosteen extract in virgin coconut oil [41]. Furthermore, nanoemulsions have the ability to fuse with and lyse bacteria, resulting in broad-spectrum antimicrobial activity [42]. However, the antibacterial activity of MSNE against *A. veronii* was significantly higher (*p* < 0.05) than that of MPE. Similarly, nano-mangosteen peel extract inhibited the growth of *Staphylococcus aureus*, *Bacillus cereus,* and *Shigella flexineri* more than the mangosteen peel extract (841 and 420 µm) [43]. Indicating a xanthone loaded in the NE or a rapid contact of the negatively charged nanodroplets with the bacterial cell wall, causing adhesion to the cell surface, membrane damage, and ultimately death [44].

The results showed that the MSNE-supplemented diet had significantly better SGR and FCR in Nile tilapia than in the control group. Similarly, ginger nanoparticles and *Aloe vera* nanoparticles significantly improved growth performance in common carp (*Cyprinus carpio*) [45] and Siberian sturgeon (*Acipenser baerii*) [46]. This can be attributed to mangosteen peel antioxidants, such as phenolic compounds, that could enhance growth [47,48]. Additionally, less than 200 nm-sized nanoemulsions had the greatest bioavailability after ingestion [49]. Therefore, supplementing mangosteen-peel-extract-loaded nanoemulsion may be another possibility for enhancing its potential role in promoting fish growth. Nile tilapia growth and health have been shown to improve following nutritional supplements [15,50], and this was also observed in the current study, despite the relatively short feeding time (30 days). However, lowering the feeding period of the MSNE diet could reduce the active compound amount and synthesized cost of MSNE. Thus, an 8-week or longer feeding trial is necessary to evaluate more substantial changes in fish growth and health.

Fish diets supplemented with nutrients may have altered immune responses due to metabolic, endocrine, or neurological pathways [51]. The mangosteen peel extract exhibited potential as an immune stimulant for fish [21]. The MSNE diet fed to Nile tilapia was observed to significantly increase serum lysozyme and ACH50 compared to the control group. The ability of fish to compete with the bacterial infection is determined by evaluating their levels of lysozyme, a key non-specific defense molecule of the immune system [52]. Lysozyme activity can be activated by an immunostimulant, which causes phagocytic cells to synthesize more lysozyme [53]. The alternative complement pathway protects fish from a wide range of possibly invading organisms [54]. In accordance with our findings, enhanced lysozyme and ACH50 were found in diets supplemented with ginger and cinnamaldehyde nanoparticles [45,51].

Nile tilapia diets fed with MSNE and MPE improved their adaptive immune response (total Ig), which is associated with the immunomodulatory activities of mangosteen peel extract [55,56]. The antigen-presenting cells of each type can uptake different sizes of particles. Macrophages are involved in the uptake of particles whose sizes range from 50 to 500 nm. Whereas dendritic-like cells are associated with the uptake, the particle size ranges from 20 to 200 nm [57,58,59]. The current study showed that the size of MSNE was 151.9 ± 1.4 nm, indicating that it can be taken up by macrophages and dendritic-like cells, which may trigger helper T cells. Furthermore, helper T cells activate B cells to differentiate into plasma cells and produce immunoglobulins [60,61]. Moreover, mangosteen peel extract has antioxidant properties that enhance the immune system and reduce oxidative stress, resulting in cell protection from oxidative stress and disease infection [16,62]. The small droplets of nanoemulsions improved the stability and absorption of mangosteen peel extract, as well as its immunostimulant and antimicrobial properties [63].

Blood biochemical analysis is one of the tools used to assess the nutritional and health status of fish [64]. In case of liver damage, ALT, AST, T-bilirubin, and D-bilirubin are released into the blood [65]. In addition, the increased levels of hepatic enzymes such as ALT and AST indicate that the fish have a high toxicity to nanoparticles. The experimental diets revealed that ALT, AST, T-bilirubin, D-bilirubin, and BUN values were significantly not different compared to the control diet, indicating that the experimental diets did not have a detrimental effect on liver and kidney function. Similarly, Nile tilapia fed *Moringa oleifera* leaf nanoparticle-supplemented diets did not increase the ALT and AST values [66].

Moreover, the fish fed MSNE showed significantly higher survival than the other groups. This suggests that the MSNE efficiency to inhibit *A. veronii* both in vitro and in vivo is due to the smaller size of the nanoparticle, which makes it easier to approach the bacterial cell wall and inhibit bacterial activity [43]. Additionally, another possibility suggests that nanoparticles can increase the solubility and absorption of herbal drugs [38]. Similar findings were reported for ginger nanoparticles and chitosan-polymer-based nanovaccines, which prevented the infection of *A. septicaemia* in *C. carpio* [45] and *A. veronii* in *Oreochromis* spp. [67]. Furthermore, we postulate that the higher survival of MSNE is also due to the several antimicrobial compounds of the mangosteen peel, such as xanthone, tannin, saponin, flavonoid, and polyphenol, which can disrupt bacterial membranes, resulting in cell hemolysis [44,68,69,70].

## 5. Conclusions

The analysis of the zeta-potential and TEM images suggested the successful preparation of nanoemulsions from mangosteen peel extract. The MSNE exhibited potent antibacterial activity. Importantly, MSNE supplementation in fish diets increased growth performance, immune parameters, and the survival rate of Nile tilapia against *A. veronii* infection. Therefore, MSNE has the potential to be employed as a supplement in sustainable Nile tilapia farming via oral administration.

## Figures and Tables

**Figure 1 animals-13-01798-f001:**
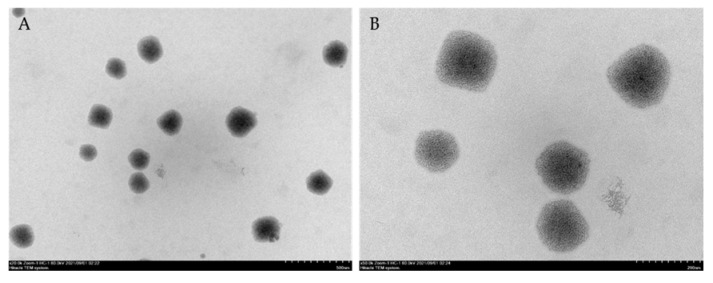
The morphology of mangosteen peel extract loaded in nanoemulsion (MSNE) observed using a transmission electron microscope. (Scale bars: 500 nm (**A**) and 200 nm (**B**)).

**Figure 2 animals-13-01798-f002:**
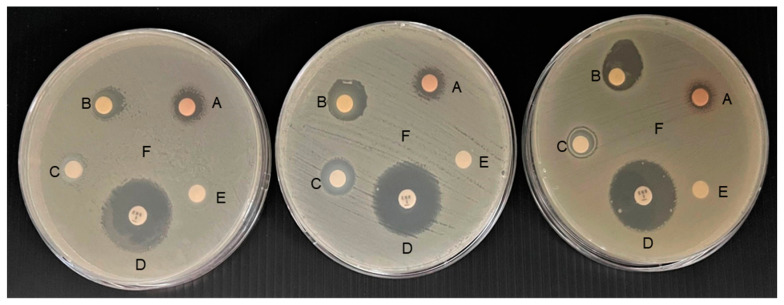
Disc diffusion assay of mangosteen peel extract (MPE) (A), mangosteen peel extract loaded in nanoemulsion (MSNE) (B), nanoemulsion (NE) (C), enrofloxacin (D), and DMSO (0.5%) (E) on *A. veronii* (F). The inhibition zone included the diameter of the disc paper (6 mm). The experiment was carried out in triplicate (*n* = 3).

**Figure 3 animals-13-01798-f003:**
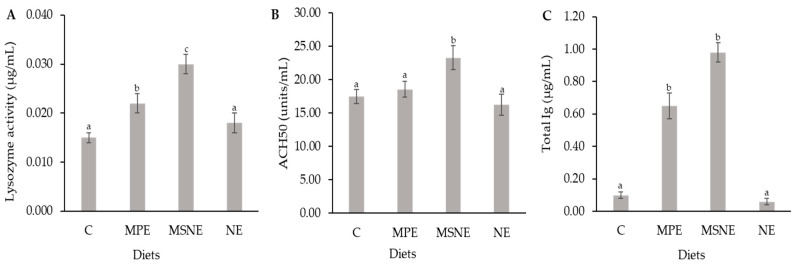
Lysozyme activity (**A**), Alternative complement hemolytic 50 (ACH50) activity (**B**) and total immunoglobulin (**C**) of Nile tilapia fed diet supplemented with Control (C), mangosteen peel extract (MPE), mangosteen peel extract loaded in nanoemulsion (MSNE), and nanoemulsion (NE) for 30 days. Data represent the mean ± SD (*n* = 6). Bars assigned with different letters indicate statistical significance (*p* < 0.05).

**Figure 4 animals-13-01798-f004:**
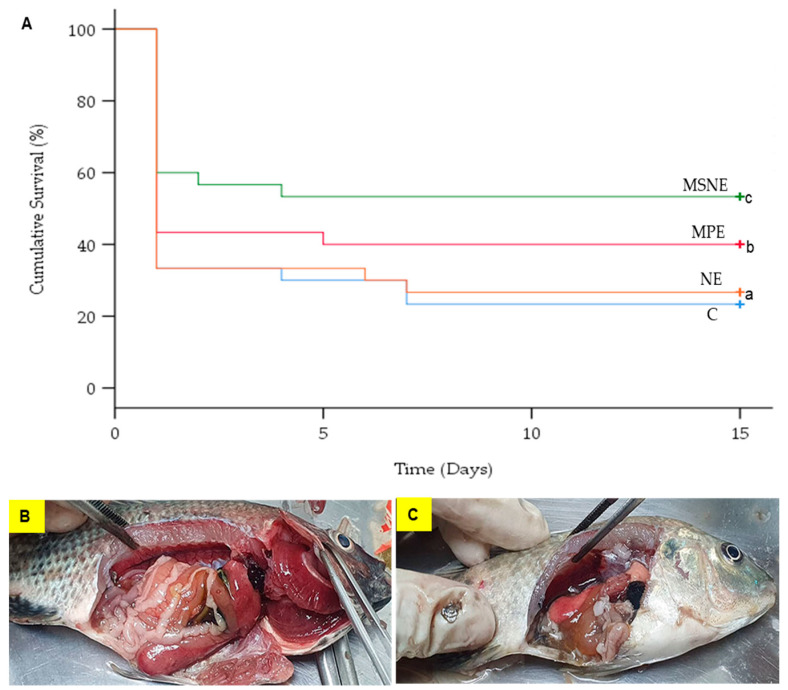
Kaplan–Meier survivorship curves over time (days) for Nile tilapia fed diet supplemented with control (C), mangosteen peel extract (MPE), mangosteen peel extract loaded in nanoemulsion (MSNE), and nanoemulsion (NE) for 30 days following *A. veronii* challenge (**A**). Dead fish exhibited necrotic gills and hemorrhage in liver (**B**), pale body surface and yellow liquid accumulation in the swollen intestine (**C**). Data represent the mean ± SD (*n* = 30). Different letters indicate statistical significance (*p* < 0.05).

**Table 1 animals-13-01798-t001:** Physicochemical properties of mangosteen peel extract loaded in nanoemulsion (MSNE) and nanoemulsion (NE).

Formulations	Average Diameter (nm)	Zeta Potential (mV)	Polydispersity Index
MSNE	151.9 ± 1.4	−38.3 ± 0.73	0.235
NE	146.4 ± 3.1	−36.4 ± 0.80	0.254

**Table 2 animals-13-01798-t002:** Antibacterial activity of mangosteen peel extract (MPE), mangosteen peel extract loaded in nanoemulsion (MSNE), and nanoemulsion (NE) on *A. veronii*.

Formulations	Inhibition Zone (mm)	MIC (µL mL^−1^)	MBC (µL mL^−1^)
MPE	11.67 ± 1.15 ^a^	62.5	62.5
MSNE	16.00 ± 3.00 ^b^	62.5	62.5
NE	10.67 ± 1.15 ^a^	62.5	125
Enrofloxacin	24.67 ± 1.15 ^c^	31.25	31.25
DMSO (0.5%)	ND	ND	ND

Results are the mean of triplicate ± SD (*n* = 3). Different superscripts in the same column show a significant difference (*p* < 0.05). MIC: minimum inhibitory concentration; MBC: minimum bactericidal concentration; ND: not detected.

**Table 3 animals-13-01798-t003:** Growth performance of Nile tilapia fed the experimental diets for 30 days.

Parameters	Control	MPE	MSNE	NE
Initial weight (g)	18.26 ± 3.37	18.70 ± 5.42	18.59 ± 6.23	18.28 ± 4.49
Final weight (g)	34.38 ± 10.53	37.07 ± 17.95	36.81 ± 15.65	35.96 ± 10.15
Weight gain (g fish^−1^)	16.12 ± 7.49 ^a^	18.37 ± 12.65 ^b^	18.22 ± 9.77 ^b^	17.68 ± 6.04 ^b^
Average daily gain (g fish^−1^ day^−1^)	0.47 ± 0.25 ^a^	0.61 ± 0.42 ^b^	0.61 ± 0.33 ^b^	0.59 ± 0.20 ^b^
Feed conversion ratio	2.16 ± 2.55 ^a^	2.02 ± 2.14 ^a^	1.75 ± 0.62 ^b^	2.00 ± 0.39 ^a^
Specific growth rate (% day^−1^)	1.64 ± 0.62 ^a^	2.00 ± 0.82 ^b^	2.18 ± 0.46 ^b^	1.92 ± 0.33 ^ab^

Results are mean ± SD (*n* = 30). Control: Basal diet; MPE: Mangosteen peel extract; MSNE: Mangosteen peel extract-loaded in nanoemulsion concentration; NE: nanoemulsion. Different superscript letters in the same row indicate statistical significance (*p* < 0.05).

**Table 4 animals-13-01798-t004:** Serum biochemical analysis of Nile tilapia fed the experimental diet for 30 days.

Parameters	Control	MPE	MSNE	NE
Blood urea nitrogen (mg dL^−1^)	4.00 ± 0.00	5.50 ± 2.12	4.00 ± 0.00	5.00 ± 1.41
Glucose (mg dL^−1^)	44.00 ± 11.31 ^a^	51.50 ± 4.95 ^a^	73.50 ± 14.72 ^b^	50.5 ± 13.23 ^a^
Alanine transferase (IU L^−1^)	3.50 ± 2.12	5.50 ± 0.71	2.50 ± 0.54	4.50 ± 1.78
Aspartate aminotransferase (IU L^−1^)	160.50 ± 8.69	180.50 ± 7.58	165.50 ± 11.92	190.50 ± 15.26
Total bilirubin (mg dL^−1^)	5.70 ± 2.69	7.25 ± 0.35	4.90 ± 0.71	6.15 ± 5.30
Direct bilirubin (mg dL^−1^)	3.47 ± 1.47	4.44 ± 0.01	3.22 ± 0.28	3.79 ± 3.09
Total cholesterol (mg dL^−1^)	147.00 ± 1.41	144.00 ± 11.31	123.50 ± 6.36	141.50 ± 23.33
Total protein (g dL^−1^)	2.35 ± 0.35 ^a^	2.15 ± 0.07 ^a^	4.00 ± 0.50 ^b^	3.30 ± 0.28 ^ab^
Albumin (g dL^−1^)	1.80 ± 0.14	1.80 ± 0.00	1.65 ± 0.07	1.95 ± 0.35

Results are mean ± SD (*n* = 6). Control: Basal diet; MPE: Mangosteen peel extract; MSNE: Mangosteen peel extract-loaded in nanoemulsion concentration; NE: nanoemulsion. Different superscript letters in a row indicate statistical significance (*p* < 0.05).

**Table 5 animals-13-01798-t005:** The cumulative mortality and relative percent survival of the experimental groups after challenge with *A. veronii*.

Formulations	Cumulative Mortality (%)	RPS (%)
Control	76.7 ± 2.2 ^a^	–
MPE	60.0 ± 3.1 ^b^	21.7 ± 2.2 ^a^
MSNE	46.7 ± 1.9 ^c^	39.1 ± 3.1 ^b^
NE	73.3 ± 4.3 ^a^	4.3 ± 0.5 ^c^

Results are the mean of triplicate ± SD (*n* = 30). Different superscripts in the same column show a significant difference (*p* < 0.05). RPS: relative percent survival, –: not applicable.

## Data Availability

Data will be made available upon request.

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
