# Peer review of "Effects of Mangosteen (Garcinia mangostana) Peel Extract Loaded in Nanoemulsion on Growth Performance, Immune Response, and Disease Resistance of Nile Tilapia (Oreochromis niloticus) against Aeromonas veronii Infection"

_animals, 2023, doi:10.3390/ani13111798_

Round 1

Reviewer 1 Report

The authors yostawonkul et al investigated mangosteen peel extract loaded nanoemulsion on growth performance, immune response and disease resistance in Nile tilapia. As far as I am concerned, the manuscript suggested that the inclusion of mangosteen peel extract loaded nanoemulsion had positive effect in nile tilapia both in growth and disease resistance. But I would suggest the authors to consider some points discussed below and revise the manuscript accordingly.

Introduction section: It’s better if you give some points about nanoemulsion. Moreover introduction needs sufficient background information about your work.

Line 89: No full form for MCT?

Some of the scientific names were not italicized. So authors check and italicize the scientific names throughout the manuscript.

Line 116: why the word starting?

Line 143: Sentence is confusing please rephrase it.

Line 145-147: Are you indicating that every 1 gram of the experimental feed contains 100 mg of extract as indicated in previous line 143?

Section 2.5: The formula 1&2 were repeated. Please check

Please check the units throughout the manuscript. For example “minutes”. In one place it was indicated as “minutes” and in another place it was given as “min”.

Table 3: The blood urea nitrogen values have been repeated twice (first and last row of the table)

Section 3.5: The authors needs to revise the section legibly for the readers. The percentage values should represent the graph Fig 4A. Here the authors used mortality percentage, survival rate percentage and relative percent of survival but the graph represent cumulative survival rate percentage. In the figure 4 legend, authors mentioned n=6 but in the experiment authors mentioned that all the thirty fish in the group received A. veronii infection then why authors considered only 6 fish for survival analysis.

In the methodology it was mentioned that the survival analysis was done for 15 days but in the graph it showed the values of 16 days.

Line 264: the value 4.3 of what?

The language is fine but need some grammatical errors 

Author Response

Response to Reviewer 1 Comments

The authors yostawonkul et al investigated mangosteen peel extract loaded nanoemulsion on growth performance, immune response and disease resistance in Nile tilapia. As far as I am concerned, the manuscript suggested that the inclusion of mangosteen peel extract loaded nanoemulsion had positive effect in nile tilapia both in growth and disease resistance. But I would suggest the authors to consider some points discussed below and revise the manuscript accordingly.

Response: Thank you for your time.

Point 1: Introduction section: It’s better if you give some points about nanoemulsion. Moreover introduction needs sufficient background information about your work.

Response: Thank you for the suggestion. We have included some points about nanoemulsion in Introduction section.

In the revised manuscript:

Lines 77–90: However, mangosteen peel extract provides low biological activity owing to its low water solubility and ease to decomposition. To overcome this challenge, nanoemulsions have been developed and used to deliver hydrophobic mangosteen peel extract. Nanoemulsion is one of the most important types of emulsion, which consists of ultra-fine particles in the range of 20–200 nm (Yostawonkul et al., 2017). Oil-in-water (o/w) nanoemulsions are a specific type of nanoemulsion that has been utilized to deliver hydrophobic compounds. Furthermore, the hydrophobic drug, which was loaded into a nanoemulsion, could improve biological activity, promote active compound stability, and increase drug absorptivity in their target organ. Nanoemulsions are typically synthesized from two immiscible solutions: oil and an aqueous solution. An appropriate amount of surfactant and energy are added to the oil-aqueous mixture to form a nanoemulsion, which can be synthesized using various techniques, such as high- and low-energy homogenization, based on its chemical composition and is easy to scale up (Calligari et al., 2016).

References

Yostawonkul, J., Surassmo, S., Iempridee, T., Pimtong, W., Suktham, K., Sajomsang, W., ... & Ruktanonchai, U. R. (2017). Surface modification of nanostructure lipid carrier (NLC) by oleoyl-quaternized-chitosan as a mucoadhesive nanocarrier. Colloids and Surfaces B: Biointerfaces, 149, 301-311.

Calligaris, S., Plazzotta, S., Bot, F., Grasselli, S., Malchiodi, A., & Anese, M. (2016). Nanoemulsion preparation by combining high pressure homogenization and high power ultrasound at low energy densities. Food Research International, 83, 25-30.

Point 2: Line 89: No full form for MCT?

Response: Thank you for the suggestion. We have included full form of MCT.

In the revised manuscript:

Line 108: Medium chain triglyceride (MCT)

Point 3: Some of the scientific names were not italicized. So authors check and italicize the scientific names throughout the manuscript.

Response: Thank you for the suggestion. We do apologize for the errors in the manuscript. The changes have been included in the revised manuscript on lines 131, 132, 144, and 145.

Point 4: Line 116: why the word starting?

Response: Thank you for the observation. We do apologize for the error in the manuscript. We have deleted the word “starting” in the revised manuscript.

Point 5: Line 143: Sentence is confusing please rephrase it.

Response: Thank you for the suggestion. We have rephrased the sentence in the revised manuscript.

Line 164: The complete randomized design with four treatments was carried out in triplicate.

Point 6: Line 145-147: Are you indicating that every 1 gram of the experimental feed contains 100 mg of extract as indicated in previous line 143?

Response: Thank you for the excellent observation. We have prepared the experimental diets from the stock solution, i.e., 100 mg/mL distilled water. In the actual experiment, 1 g of feed contained 6.25 mg of MPE and MSNE, respectively. To avoid further confusion regarding NE solution, we have converted extract contents into µL per mL. In the revised manuscript, we have added per gram of extract concentration in the MPE and MSNE groups. However, we welcome further suggestions from the reviewer.

In the revised manuscript

Lines 166-167: The treatment diets were as follows: a control diet (Control), MPE (62.5 µL/mL, i.e., 6.25 mg/1 g of feed), MSNE (62.5 µL/mL, i.e., 6.25 mg/1 g of feed), and NE (62.5 µL/mL).

Point 7: Section 2.5: The formula 1&2 were repeated. Please check

Response: Thank you for the observation. We do apologize for the error in the manuscript. Furthermore, we have deleted the repeated formula in the revised manuscript.

Point 8: Please check the units throughout the manuscript. For example “minutes”. In one place it was indicated as “minutes” and in another place it was given as “min”.

Response: Thank you for the suggestion. We have changed minutes to min in the revised manuscript.

Point 9: Table 3: The blood urea nitrogen values have been repeated twice (first and last row of the table)

Response: Thank you for the suggestion. We have deleted the repeated blood urea nitrogen values in the revised manuscript.

Point 10: Section 3.5: The authors needs to revise the section legibly for the readers. The percentage values should represent the graph Fig 4A. Here the authors used mortality percentage, survival rate percentage and relative percent of survival but the graph represent cumulative survival rate percentage. In the figure 4 legend, authors mentioned n=6 but in the experiment authors mentioned that all the thirty fish in the group received A. veronii infection then why authors considered only 6 fish for survival analysis.

Response: Thank you for the suggestion. The Kaplan-Meier method is used to analyze the survival of challenged fish with the Log Rank (Mentel-Cox) test, which tests the null hypothesis that there is no difference in the overall survival distributions between the groups in the population. A log rank test showed that the survival percentages of the four experimental groups were significantly different (X2(3) = 6.893, p < 0.05) (Fig. 4A).

Figure 4. Kaplan–Meier survivorship curves over time (days) for Nile tilapia fed diet supplemented with control (C), mangosteen peel extract (MPE), mangosteen peel extract loaded in nanoemulsion (MSNE), and nanoemulsion (NE) for 30 days following A. veronii challenge (A).

Furthermore, we have included a table for cumulative mortality and relative percent survival of the experimental groups and updated it in the revised manuscript (Line 220).

Table 4. The cumulative mortality and relative percent survival of the experimental groups after challenged with A. veronii

Formulations

Cumulative mortality (%)

RPS (%)

Control

76.7±2.2a

MPE

60.0±3.1b

21.7±2.2a

MSNE

46.7±1.9c

39.1±3.1b

NE

73.3±4.3a

4.3±0.5c

Results are the mean of triplicate ± SD (n=3). Different superscripts in the same column show a significant difference (p < 0.05). RPS: relative percent survival, –: not applicable.

Importantly, we would like to inform you that it was a typing error in the figure 4 legend, which mentioned n = 6. We further assured a reviewer that it is impossible to use only six fish for survival analysis. We apologize for our mistake. We have corrected n = 30 in the figure 4 legend of the revised manuscript.

Point 11: In the methodology it was mentioned that the survival analysis was done for 15 days but in the graph it showed the values of 16 days.

Response: Thank you for the excellent observation. It was an error while selecting the chart data range for horizontal axis labels in Excel, which showed the values for 16 days instead of 15 days. We apologize for our mistake.

Point 12: Line 264: the value 4.3 of what?

Response: Thank you for the excellent observation. However, 4.3 was the RPS value of NE-supplemented diets fed Nile tilapia. We have corrected it in the revised manuscript.

Point 13: Comments on the Quality of English Language: The language is fine but need some grammatical errors 

Response: Thank you. We have corrected the grammatical errors in the revised manuscript.

Reviewer 2 Report

The study explored the effects of mangosteen peel extract loaded in nanoemulsion (MSNE) on inhibition of Aeromonas veronii (in vitro) and growth performance, serum biochemical parameters, immune response, and disease resistance of Nile tilapia. The results are interesting and potentially applicable in sustainable tilapia aquaculture. However, there are some concerns that need to be addressed:1. The feeding trial period (30 days) is too short. An 8-week or longer feeding trial is typically needed to investigate the effects of dietary supplements on fish growth and health.2. Water quality parameters such as ammonia nitrogen should be provided in the Materials and Methods section. These environmental factors can significantly affect the experimental results.3. The reference format is inconsistent. For example, the case of the article titles is not uniform. The references should follow the journal's style guide.
In summary, this study provides interesting results on the effects of MSNE on Nile tilapia. However, additional details on experimental conditions and data analysis are needed to improve the reproducibility and rigor of this study. Addressing these concerns will strengthen the quality and credibility of this paper.

Good.

Author Response

Response to Reviewer 2 Comments

The study explored the effects of mangosteen peel extract loaded in nanoemulsion (MSNE) on inhibition of Aeromonas veronii (in vitro) and growth performance, serum biochemical parameters, immune response, and disease resistance of Nile tilapia. The results are interesting and potentially applicable in sustainable tilapia aquaculture. However, there are some concerns that need to be addressed:

Response: Thank you for your time.

Point 1: The feeding trial period (30 days) is too short. An 8-week or longer feeding trial is typically needed to investigate the effects of dietary supplements on fish growth and health.

Response: Thank you for your suggestion. We have agreed with you that an 8-week or longer feeding trial could provide more significant differences in terms of fish growth and health. However, lowering the feeding period of the MSNE diet could reduce the active compound amount and synthesized cost of MSNE. Moreover, in our experiments, the difference between the control diet and the MSNE diet could be observed during the short feeding period (30 days). In addition, previous studies have shown that a feeding trial period of 30 days can investigate the effects of dietary supplements on fish growth and health (Abaho et al., 2022; Sewaka et al., 2019).

References

Abaho I, Masembe C, Akoll P, Jones CLW. The use of plant extracts to control tilapia reproduction: Current status and future perspectives. Journal of the World Aquaculture Society. 2022;53(3):593-619.

Sewaka, M.; Trullas, C.; Chotiko, A.; Rodkhum, C.; Chansue, N.; Boonanuntanasarn, S.; Pirarat, N. Efficacy of synbiotic Jerusalem artichoke and Lactobacillus rhamnosus GG-supplemented diets on growth performance, serum biochemical parameters, intestinal morphology, immune parameters and protection against Aeromonas veronii in juvenile red tilapia (Oreochromis spp.). Fish & Shellfish Immunology 2019, 86, 260-268.

Point 2: Water quality parameters such as ammonia nitrogen should be provided in the Materials and Methods section. These environmental factors can significantly affect the experimental results.

Response: Thank you for the suggestion. We have included Ammonia nitrogen in the revised manuscript.

Lines 157-159: Ammonia nitrogen was managed by exchanging 50% of the water every two days and measuring the ammonia nitrogen using test kits (V-Unique, USA), which indicated less than 0.02 mg/L.

Point 3: The reference format is inconsistent. For example, the case of the article titles is not uniform. The references should follow the journal's style guide.

Response: Thank you for the excellent suggestion. We have checked and improved the article titles of the references in the revised manuscript.

Point 4: In summary, this study provides interesting results on the effects of MSNE on Nile tilapia. However, additional details on experimental conditions and data analysis are needed to improve the reproducibility and rigor of this study. Addressing these concerns will strengthen the quality and credibility of this paper.

Response: Thank you for the suggestion. The additional details on the experimental conditions were included in the revised manuscript. Furthermore, survival data was analyzed using the Kaplan-Meier method with the log rank (Mantel-Cox) test to determine the significant differences.

In the revised manuscript

Lines 153-154: Three hundred sixty monosex (male) tilapia (15.35±0.91 g) were obtained from the Napho Phanpla Limited Partnership tilapia farm in Nakhon Si Thammarat, Thailand

Lines 157-159: Ammonia nitrogen was managed by exchanging 50% of the water every two days and measuring the ammonia nitrogen using test kits (V-Unique, USA), which indicated less than 0.02 mg/L.

Line 164: The complete randomized design with four treatments was carried out in triplicate

Lines 166-167: MPE (62.5 µL/mL, i.e., 6.25 mg/1 g of feed), MSNE (62.5 µL/mL, i.e., 6.25 mg/1 g of feed).

Lines 219-220: The cumulative survival percentages of the experimental groups were analyzed using the Kaplan-Meier method and the Log Rank (Mantel-Cox) test.

Point 5: Comments on the Quality of English Language: Good.

Response: Thank you for the appreciation.

Reviewer 3 Report

The MS titled “Effects of Mangosteen (Garcinia mangostana) Peel Extract Loaded in Nanoemulsion on Growth Performance, Immune Response, and Disease Resistance of Nile Tilapia (Oreochromis niloticus) against Aeromonas veronii infection” was well-conducted. And the manuscript is generally well-written. However, there are some issues which need to be clarified before publication:

1.      “in vitro” should be in italics

2.      “In conclusion, mangosteen peel extract loaded in nanoemulsion has the potential to be used as supplements in Nile tilapia culture.” Please clarify that if the mangosteen peel extract is added into diets or water.

3.      Line 49, please introduce the production of Nile tilapia (Oreochromis niloticus) at present.

4.      Line 52-53, please make a broader introduction of the disease threats in tilapia farming, not limited to only Aeromonas veronii.

5.      Line 61-69, please include the analysis of some bioactive compounds for this study. And add the results in the M&M part.

6.      The advantage and disadvantage of nano-dietary delivery method should be introduced in more details.

7.      Line 86-94, please add the aforementioned some bioactive compounds information.

8.      Line 141, “in vivo” should be in “italics”

9.      “fat” should be changed into “lipid”

10.   Specific growth rate expresses similar meanings of weight gain. Maybe one of them is enough.

11.   Regarding the serum alanine transaminase (ALT) and aspartate aminotransferase (AST), although many fish papers tested these parameters, I don’t see any value of them. These parameters are from human clinical diagnosis. However, in human clinical diagnosis, there is a normal range of each parameter based on big data survey, so that from a medical examination report we can know a certain parameter is too low or too high compared to the normal range. In fish, we don’t have this normal range of each parameter. Thus, how can we know a value is too high or too low? I don’t think low values are always good. There should be a normal range for our judgement, however, we don’t have that criteria. Please delete these parameters.

12.   Line 187, “2.8. Challemged study”, please correct the grammar error

Please check the grammars. 

Author Response

Response to Reviewer 3 Comments

The MS titled “Effects of Mangosteen (Garcinia mangostana) Peel Extract Loaded in Nanoemulsion on Growth Performance, Immune Response, and Disease Resistance of Nile Tilapia (Oreochromis niloticus) against Aeromonas veronii infection” was well-conducted. And the manuscript is generally well-written. However, there are some issues which need to be clarified before publication:

Response: Thank you for your time.

Point 1: in vitro” should be in italics

Response: Thank you for the suggestion. We have changed "in vitro" to italics in the revised manuscript.

Point 2: “In conclusion, mangosteen peel extract loaded in nanoemulsion has the potential to be used as supplements in Nile tilapia culture.” Please clarify that if the mangosteen peel extract is added into diets or water.

Response: Thank you for your suggestion. In this research, MPE, MSNE, and NE were synthesized and added into fish diets to use as feed supplement in Nile tilapia via oral administration. Furthermore, the control diet was prepared by adding distilled water in commercial feed.  

In the revised manuscript

Lines 372-377: The analysis of zeta-potential and TEM images suggested the successful preparation of nanoemulsions from mangosteen peel extract. The MSNE exhibited potent antibacterial activity. Importantly, MSNE supplementation in fish diets increased growth performance, immune parameters, and the survival rate of Nile tilapia against A. veronii infection. Therefore, MSNE has the potential to be employed as a supplement in sustainable Nile tilapia farming via oral administration.

Point 3: Line 49, please introduce the production of Nile tilapia (Oreochromis niloticus) at present.

Response: Thank you for the suggestion. We have included the production data of Nile tilapia in the revised manuscript.

Lines 50-52: The global production of Nile tilapia in 2020 was 4,407.2 thousand tons (FAO 2022), with Thailand contributing 210,419 tons with a 2.2% increase in 2021 (Fisheries Do, 2021).

References

FAO. The state of world fisheries and aquaculture 2022- Towards blue transformation; FAO: Rome, 2022; p. 266.

Fisheries Do. 2021. Fisheries Statistics of Thailand 2021. National Inland Fisheries Institute, Department of Fisheries, Bangkok.

Point 4: Line 52-53, please make a broader introduction of the disease threats in tilapia farming, not limited to only Aeromonas veronii.

Response: Thank you for the excellent suggestion. We have added broader introduction of the disease threats in tilapia farming in the revised manuscript.

Lines 53-57: Natural pathogen infections in tilapia have been found, such as those caused by Aeromonas spp., Flavobacterium columnare, Edwardsiella spp., Francisella sp., and Streptococcus agalactiae. Among bacterial diseases, Aeromonas veronii has recently resulted in high mortality rates on Thai tilapia farms (Dong et al., 2015 and 2017).

Reference

Dong, H.T.; Nguyen, V.V.; Le, H.D.; Sangsuriya, P.; Jitrakorn, S.; Saksmerprome, V.; Senapin, S.; Rodkhum, C. Naturally concurrent infections of bacterial and viral pathogens in disease outbreaks in cultured Nile tilapia (Oreochromis niloticus) farms. Aquaculture 2015, 448, 427-435.

Dong, H.T.; Techatanakitarnan, C.; Jindakittikul, P.; Thaiprayoon, A.; Taengphu, S.; Charoensapsri, W.; Khunrae, P.; Rattanarojpong, T.; Senapin, S. Aeromonas jandaei and Aeromonas veronii caused disease and mortality in Nile tilapia, Oreochromis niloticus (L.). Journal of Fish Diseases 2017, 40, 1395-1403.

Point 5: Line 61-69, please include the analysis of some bioactive compounds for this study. And add the results in the M&M part.

Response: Thank you for the excellent suggestion. We have included the bioactive compounds in the revised manuscript.

Lines 102-104: The ethanol extract of the mangosteen peels is composed of xanthones (Yoshimura et al., 2015; Kusmayadi et al., 2018), which possess antimicrobial, antioxidant, anti-inflammatory, and anti-proliferative activities (Ansori et al., 2020).

References

Ansori, A.N.M.; Fadholly, A.; Hayaza, S.; Susilo, R.J.K.; Inayatillah, B.; Winarni, D.; Husen, S.A. A review on medicinal properties of mangosteen (Garcinia mangostana L.). Research Journal of Pharmacy and Technology 2020, 13, 974-982.

Kusmayadi A, Adriani L, Abun A, Muchtaridi M, Tanuwiria UH. The effect of solvents and extraction time on total xanthone and antioxidant yields of mangosteen peel (Garcinia mangostana L.) extract. Drug Invention Today. 2018;10:2572-6.

Yoshimura M, Ninomiya K, Tagashira Y, Maejima K, Yoshida T, Amakura Y. Polyphenolic Constituents of the Pericarp of Mangosteen (Garcinia mangostana L.). Journal of Agricultural and Food Chemistry. 2015;63(35):7670-4.

Point 6: The advantage and disadvantage of nano-dietary delivery method should be introduced in more details.

Response: Thank you for the suggestion. We have included some points about nanoemulsion in Introduction section.

In the revised manuscript:

Lines 77–90: However, mangosteen peel extract provides low biological activity owing to its low water solubility and ease to decomposition. To overcome this challenge, nanoemulsions have been developed and used to deliver hydrophobic mangosteen peel extract. Nanoemulsion is one of the most important types of emulsion, which consists of ultra-fine particles in the range of 20–200 nm (Yostawonkul et al., 2017). Oil-in-water (o/w) nanoemulsions are a specific type of nanoemulsion that has been utilized to deliver hydrophobic compounds. Furthermore, the hydrophobic drug, which was loaded into a nanoemulsion, could improve biological activity, promote active compound stability, and increase drug absorptivity in their target organ. Nanoemulsions are typically synthesized from two immiscible solutions: oil and an aqueous solution. An appropriate amount of surfactant and energy are added to the oil-aqueous mixture to form a nanoemulsion, which can be synthesized using various techniques, such as high- and low-energy homogenization, based on its chemical composition and is easy to scale up (Calligari et al., 2016).

References

Yostawonkul, J., Surassmo, S., Iempridee, T., Pimtong, W., Suktham, K., Sajomsang, W., ... & Ruktanonchai, U. R. (2017). Surface modification of nanostructure lipid carrier (NLC) by oleoyl-quaternized-chitosan as a mucoadhesive nanocarrier. Colloids and Surfaces B: Biointerfaces, 149, 301-311.

Calligaris, S., Plazzotta, S., Bot, F., Grasselli, S., Malchiodi, A., & Anese, M. (2016). Nanoemulsion preparation by combining high pressure homogenization and high power ultrasound at low energy densities. Food Research International, 83, 25-30.

Point 7: Line 86-94, please add the aforementioned some bioactive compounds information.

Response: Thank you for the suggestion. The ethanol extract of the mangosteen peels is composed of xanthones (Yoshimura 2015, Kusmayadi 2018), which possess antimicrobial, antioxidant, anti-inflammatory, and anti-proliferative activities (Ansori, 2020).

References

Ansori, A.N.M.; Fadholly, A.; Hayaza, S.; Susilo, R.J.K.; Inayatillah, B.; Winarni, D.; Husen, S.A. A review on medicinal properties of mangosteen (Garcinia mangostana L.). Research Journal of Pharmacy and Technology 2020, 13, 974-982.

Kusmayadi A, Adriani L, Abun A, Muchtaridi M, Tanuwiria UH. The effect of solvents and extraction time on total xanthone and antioxidant yields of mangosteen peel (Garcinia mangostana L.) extract. Drug Invention Today. 2018;10:2572-6.

Yoshimura M, Ninomiya K, Tagashira Y, Maejima K, Yoshida T, Amakura Y. Polyphenolic Constituents of the Pericarp of Mangosteen (Garcinia mangostana L.). Journal of Agricultural and Food Chemistry. 2015;63(35):7670-4.

Point 8: Line 141, “in vivo” should be in “italics”

Response: Thank you for the suggestion. We have changed “in vivo” to “italics” in the revised manuscript.

Point 9:  “fat” should be changed into “lipid”

Response: Thank you for the suggestion. We have changed “fat” to “lipid” in the revised manuscript.

Line 172: Changed “fat” into “lipid”

Point 10:  Specific growth rate expresses similar meanings of weight gain. Maybe one of them is enough.

Response: Thank you for the suggestion. We agreed with a reviewer that a specific growth rate expresses similar meanings for weight gain. However, in our study, weight gain indicates grams per fish at the end of the experiment, and specific growth rate expresses the percentage increase in fish weight per day. Therefore, we would like to keep both parameters in the revised manuscript.

Point 11: Regarding the serum alanine transaminase (ALT) and aspartate aminotransferase (AST), although many fish papers tested these parameters, I don’t see any value of them. These parameters are from human clinical diagnosis. However, in human clinical diagnosis, there is a normal range of each parameter based on big data survey, so that from a medical examination report we can know a certain parameter is too low or too high compared to the normal range. In fish, we don’t have this normal range of each parameter. Thus, how can we know a value is too high or too low? I don’t think low values are always good. There should be a normal range for our judgement, however, we don’t have that criteria. Please delete these parameters.

Response: Thank you for the excellent explanation on the ALT and AST, and we agreed with the reviewers. Furthermore, in our study, we compared the ALT and AST values of the experimental diet with those of the control diet. The results showed that ALT and AST were not different from the control diet, indicating that the experimental diet had no effects on liver and kidney function. Therefore, we would like to keep these parameters in the revised manuscript.

Point 12: Line 187, “2.8. Challemged study”, please correct the grammar error

Response: Thank you for the suggestion. We have changed “Challemged study” to “Challenged study” in the revised manuscript.

Point 13: Comments on the Quality of English Language: Please check the grammars. 

Response: Thank you for the suggestion. We have checked and improved the grammars in the revised manuscript.

Round 2

Reviewer 3 Report

No  further comments.

Author Response

Response to Reviewer 2 Comments

Point 1: No further comments.

Response: Thank you for your time.
